# Extensive hydrogen incorporation is not necessary for superconductivity in topotactically reduced nickelates

Purnima P. Balakrishnan [1,15] ✉, Dan Ferenc Segedin [2,15], Lin Er Chow [3,15],
P. Quarterman [1], Shin Muramoto[4], Mythili Surendran [5,6], Ranjan K. Patel[7],
Harrison LaBollita [8], Grace A. Pan [2], Qi Song [2], Yang Zhang [9],
Ismail El Baggari[9], Koushik Jagadish[5], Yu-Tsun Shao [5,10],
Berit H. Goodge [11,12,13], Lena F. Kourkoutis[11,12], Srimanta Middey [7],
Antia S. Botana [8], Jayakanth Ravichandran [5,6,14] ✉, A. Ariando [3] ✉,
Julia A. Mundy [2] ✉ & Alexander J. Grutter [1] ✉

A key open question in the study of layered superconducting nickelate films is the role that hydrogen incorporation into the lattice plays in the appearance of the superconducting state. Due to the challenges of stabilizing highly crystalline square planar nickelate films, films are prepared by the deposition of a more stable parent compound which is then transformed into the target phase *via* a topotactic reaction with a strongly reducing agent such as $CaH_2$. Recent studies, both experimental and theoretical, have introduced the possibility that the incorporation of hydrogen from the reducing agent into the nickelate lattice may be critical for the superconductivity. In this work, we use secondary ion mass spectrometry to examine superconducting $La_{1-x}X_xNiO_2$ / $SrTiO_3$ ($X$ = Ca and Sr) and $Nd_6Ni_5O_{12}$ / $NdGaO_3$ films, along with non-superconducting $NdNiO_2$ / $SrTiO_3$ and $(Nd,Sr)NiO_2$ / $SrTiO_3$. We find no evidence for extensive hydrogen incorporation across a broad range of samples, including both superconducting and non-superconducting films. Theoretical calculations indicate that hydrogen incorporation is broadly energetically unfavorable in these systems, supporting our conclusion that extensive hydrogen incorporation is not generally required to achieve a superconducting state in layered square-planar nickelates.

Superconductivity in nickelates has been pursued ever since the discovery of the cuprates[1–5], but it was not until 2019 that it was demonstrated in thin films of the infinite-layer compound $NdNiO_2$ via hole doping with Sr[6]. This discovery introduced a novel family of layered nickelate superconductors that has now been extended to include the Pr- and La- analogs of the infinite-layer compound as well as the five-layer material $Nd_6Ni_5O_{12}$[7–12]. While superconducting nickelates exhibit many interesting phenomena[13–17], they also represent a unique materials synthesis challenge[18–22]. In general, layered square-planar nickelates cannot be synthesized directly, instead requiring a two-step fabrication method wherein an oxygen-rich precursor material is grown by traditional thin film deposition methods and then topotactically reduced, as illustrated in Fig. 1a, b[23]. Typically, the reduction is performed *via* a thermal anneal employing a chemical reducing agent and oxygen sink, such as $H_2$, NaH, or $CaH_2$[6,24–26]. One of the most pressing open questions is the degree to which the reduction process

mundy@fas.harvard.edu; alexander.grutter@nist.gov

**Fig. 1 | Representation of materials and methods used in this study.** Schematic crystal structures for precursor phase and reduced **a** $n = \infty$ and **b** $n = 5$ layered square-planar nickelate compounds. **c** Schematic of the ToF-SIMS measurement technique. **d** SIMS spectra measured separately for positive and negative ions are analyzed by identifying peaks by mass-to-charge ratio ($m/z$) and extracting integrated area.

incorporates hydrogen into the nickelate film, and whether hydrogen is important in stabilizing superconductivity.

A notable recent study by Ding et al. reported that hydrogen is critical for the emergence of superconductivity, requiring a stoichiometry around $Nd_{0.8}Sr_{0.2}NiO_2H_{0.25}$[27]. However, in this study, hydrogen and oxygen stoichiometry are highly correlated through reduction. Furthermore, recent results using alternative reduction processes with no hydrogen source have cast doubt on the role of incorporated hydrogen on the electronic state[9,25]. Previous theoretical works have argued that $RNiO_2$ ($R = $ La, Nd) could be energetically unstable with respect to topotactic hydrogen, significantly altering the electronic structure[28–30]. In light of Ref. 27, some calculations have shown that an optimal H concentration may be beneficial to promote superconductivity[31], while others have indicated that the electron phonon-coupling in hydrogen-intercalated nickelates is not strong enough to drive electron pairing and thus cannot be responsible for the superconductivity[32].

Given the differing conclusions in the literature, a comprehensive examination of the role of hydrogen incorporation in superconducting nickelates is urgently needed. To understand more broadly applicable trends rather than the specifics of one sample type or fabrication protocol, we used time-of-flight secondary ion mass spectrometry (ToF-SIMS) to study the relationship between hydrogen incorporation and superconductivity in a broad range of nickelate films grown and reduced by three different research groups. The films used in this study were grown *via* either molecular beam epitaxy (MBE) or pulsed laser deposition (PLD), in a variety of geometries, and reduced with $CaH_2$ at different conditions. As the energetics of incorporating hydrogen may vary greatly depending on stoichiometry and

structure[28], we compared multiple nickelate systems, including superconducting examples of $La_{1-x}Ca_xNiO_2$, $La_{1-x}Sr_xNiO_2$, and $Nd_6Ni_5O_{12}$. We also examined non-superconducting $NdNiO_2$ and $Nd_{1-x}Sr_xNiO_2$. We find no evidence that a large concentration of incorporated hydrogen is necessary to observe superconductivity. Instead, a wide range of films, superconducting and non-superconducting, exhibit $H^-$ intensities that are similar to the substrate background. Theoretical calculations support this picture, revealing that hydrogen incorporation is energetically unfavorable across all materials studied in this work.

## Results

As illustrated in Fig. 1c, ToF-SIMS is a destructive technique in which an ion beam sputters through the film, and ejected molecular ions are analyzed using a mass spectrometer to provide a depth- and element-resolved picture of the ejected species and, thereby, chemical composition. ToF-SIMS allows the isolation and identification of elemental $H^{\pm}$, and $O^{\pm}$ as well as larger ejected molecules such as $OH^{\pm}$, $O_2^{\pm}$, and $TiO_2^{\pm}$, as shown in Fig. 1d. The change in molecular species intensity over time as the sample is sputtered results in a depth-resolved understanding of the chemical composition with depth, in which layers only a few nm thick may be readily separated.

However, the measured intensity depends significantly on the sputtering conditions, chemical environment, film composition, density, electronic state, and prevalence of structural defects. Absolute scaling of stoichiometry and depth, therefore, requires calibration standards with known stoichiometry and a similar chemical environment to the films of interest. Since the defect levels and chemistry in superconducting nickelates evolve extensively during the reduction

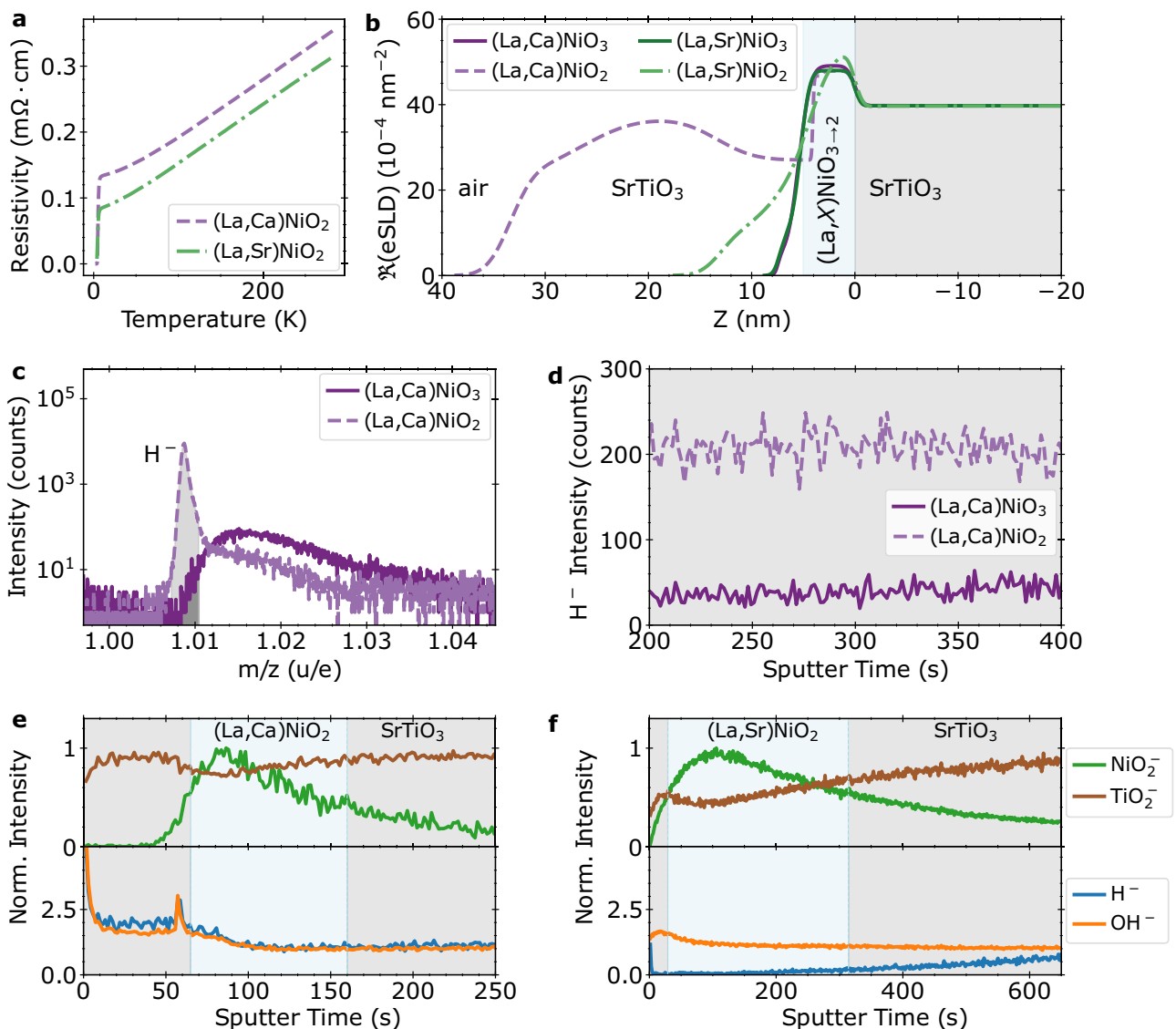

**Fig. 2 | Characterization of superconducting La$_x$(Sr,Ca)$_{1-x}$NiO$_2$ films.**
**a** Resistivity vs. temperature for the superconducting La$_{0.78}$Ca$_{0.22}$NiO$_2$ and La$_{0.8}$Sr$_{0.2}$NiO$_2$ samples, showing a clear transition and large RRR. **b** XRR depth profiles of the as-grown and superconducting films, specifically the real component of the scattering length density (SLD). **c** Intensity vs. mass-to-charge ratio near the H$^-$ peak for an as-grown La$_{0.78}$Ca$_{0.22}$NiO$_3$ and superconducting La$_{0.78}$Ca$_{0.22}$NiO$_2$

film, integrated over the entire measurement time. **d** Raw intensity (counts) of the H$^-$ peak in the substrate region for the same as-grown La$_{0.78}$Ca$_{0.22}$NiO$_3$ and superconducting La$_{0.78}$Ca$_{0.22}$NiO$_2$. **e** SIMS depth profile of superconducting La$_{0.78}$Ca$_{0.22}$NiO$_2$. **f** SIMS depth profile of superconducting La$_{0.8}$Sr$_{0.2}$NiO$_2$. Note that the La$_{0.8}$Sr$_{0.2}$NiO$_2$ film was sputtered at a lower ion beam energy than the La$_{0.78}$Ca$_{0.22}$NiO$_2$ due to the thinner cap layer.

process, such standards are nearly impossible to obtain, and we instead adopt the convention of Ding et al., in which the hydrogen level observed in the SrTiO$_3$ or NdGaO$_3$ substrate is considered to be the background level representing minimal hydrogen[27].

**Superconducting La$_{1-x}$(Sr,Ca)$_x$NiO$_2$**
We first present results from two doped superconducting infinite-layer systems: La$_{0.78}$Ca$_{0.22}$NiO$_2$ and La$_{0.8}$Sr$_{0.2}$NiO$_2$ grown by pulsed laser deposition on SrTiO$_3$ substrates, as described in the Methods section. The quality of representative samples has been previously demonstrated through X-ray diffraction (XRD), cross-sectional scanning transmission electron microscopy (STEM), and electron energy loss spectroscopy (EELS) analysis[33,34]. To ensure depth-wise uniformity, the film thickness was limited to below 6 nm. Figure 2a shows the superconducting transitions for these samples, which have a residual resistivity ratio ≈4.1, comparable to the highest reported values so far[21,35].

After reduction, an amorphous SrTiO$_3$ cap is deposited to act as an oxidation barrier, with varying thickness due to the challenges associated with room-temperature growth. Film and cap thicknesses were verified using X-ray reflectometry (XRR), shown in Fig. 2b, which reveals that the initial perovskite phases are uniform with the expected scattering length densities. After reduction, the sharp interfaces slightly roughen, likely linked to the energetic deposition of the caps. While here we focus on superconducting films, we also measured the as-grown perovskite film from the same growth, the details of which can be found in the Supplementary Information.

To understand the sensitivity of this experiment to hydrogen, we first compare the overall hydrogen content of as-grown La$_{0.78}$Ca$_{0.22}$NiO$_3$ and superconducting La$_{0.78}$Ca$_{0.22}$NiO$_2$ samples in Fig. 2c. Here, we show the H$^-$ peak for both samples integrated across all sputtering times. The as-grown sample contains negligible hydrogen within either the film or substrate, indicating an extremely clean

growth and handling process. The superconducting sample, in contrast, exhibits a clearly-resolved $H^-$ peak much larger than the measurement background. The additional hydrogen introduced into the reduced superconducting sample is easily detectable. We next compare the integrated peak intensity, indicated by the shaded region, over sputter time (depth). For this same pair of samples, Fig. 2d shows the $H^-$ intensity in just the $SrTiO_3$ substrate. Interestingly, while the substrate intensity of the as-grown sample is less than 40 counts per frame, the $SrTiO_3$ substrate associated with the superconducting $La_{0.78}Ca_{0.22}NiO_2$ is an excellent match for the samples in Ding et al.[27], with approximately 200 counts per frame of $H^-$. Thus we may be confident that the observed hydrogen levels in the substrates are above the instrumental detection limit and closely match previous observations.

Having firmly established that the hydrogen levels reported previously in superconducting films are readily detectable with the instrument used in this study, we show SIMS data from superconducting $La_{0.78}Ca_{0.22}NiO_2$ and $La_{0.8}Sr_{0.2}NiO_2$ in Fig. 2e, f. Here the $NiO_2^-$ and $TiO_2^-$ intensities are normalized to their maximum values while the $H^-$ and $OH^-$ intensities are normalized to the steady-state value within the substrate; alternative normalizations and raw counts are shown in Supplementary Note 5. The film and substrate positions are indicated by the peak and dip in $NiO_2^-$ and $TiO_2^-$ intensities, respectively. The trends in $H^-$ and $OH^-$ intensities clearly disagree with prior reports: the superconducting $La_{0.78}Ca_{0.22}NiO_2$ and $La_{0.8}Sr_{0.2}NiO_2$ films do not exhibit the large 1–3 order of magnitude increases in $H^-$ or $OH^-$ signal which would be expected for extensive, multiple-percent hydrogen incorporation[27,36–38].

Instead, apart from the quickly decaying signal associated with surface adsorbates, the $H^-$ and $OH^-$ signals within the $La_{0.78}Ca_{0.22}NiO_2$ film are invariant within a factor of two of the signals within the substrate. Interestingly, the $La_{0.78}Ca_{0.22}NiO_2$ sample, with a thicker amorphous $SrTiO_3$ cap (29 nm), exhibits higher $H^-$ intensity within the cap than within the nickelate film, concentrated near the interface. In the $La_{0.8}Sr_{0.2}NiO_2$ sample with a thinner $SrTiO_3$ cap (approximately 6 nm), $H^-$ is much lower in the nickelate film than either the $SrTiO_3$ substrate or the other superconducting film. We speculate that the $SrTiO_3$ cap may play a role in hydrogen capture or transport[39]. Most importantly, the coexistence of different low hydrogen concentrations with superconductivity definitively demonstrates that extensive hydrogen doping is not required for superconductivity in the infinite-layer nickelates.

## Superconducting $Nd_6Ni_5O_{12}$

To test whether our findings are applicable more broadly within the square-planar nickelate family, beyond the infinite-layer structure, we examine the superconducting quintuple-layer nickelate $Nd_6Ni_5O_{12}$. This film consists of 23 nm $Nd_6Ni_5O_{12}$ synthesized on $NdGaO_3$ (110) (see Synthesis Section 2 for details), with 10 nm titanium followed by 100 nm platinum patterned on the film surface as electrodes. Figure 3a shows the zero-field superconducting transition of this sample from Ref. 10, with a residual resistivity ratio of 3.8. Further characterization of this sample can be found in Ref. 10. Figure 3b shows a representative STEM image of this sample, revealing the five-layer square-planar structure.

Figure 3c plots the SIMS depth profile of this superconducting sample. $NiO_2^-$ and $GaO_2^-$ peaks are normalized to their maximum values, and clearly identify the electrode, film, and substrate regions. As before, we obtain information regarding the hydrogen concentration by examining the relative intensity of the $H^-$ and $OH^-$ peaks in the film and substrate. An advantage of the relatively thick electrode is the removal of surface contaminant effects from the measurement. Both the $H^-$ and $OH^-$ intensities are low in the platinum and titanium, increase slowly in the $Nd_6Ni_5O_{12}$ film, and further increase deeper into the $NdGaO_3$ substrate. Similar to the superconducting $La_{0.8}Sr_{0.2}NiO_2$

sample, we find that the hydrogen content appears to be highest in the substrate, although again the nickelate film and substrate intensities are very similar. Once again there is no evidence of an order of magnitude increase in $H^-$ intensity in the film.

## Non-superconducting films

With little evidence of extensive hydrogen incorporation in high-quality superconducting samples, the question remains whether some structures or processes are more susceptible to hydrogen. We speculate that films with increased defect densities, whether due to growth conditions or from long or overly aggressive reduction treatments, may incorporate additional hydrogen as a defect compensation mechanism. These films do not exhibit superconductivity, but do provide a mechanism for understanding the extent to which hydrogen can be incorporated during reduction and whether it might inhibit the fabrication of superconducting films.

We first consider 17 nm $NdNiO_3$/$SrTiO_3$ (001) films grown by MBE and subjected to an incomplete reduction, at a lower temperature but for longer times compared to the optimized treatment for achieving high-quality $NdNiO_2$. XRD scans shown in Fig. 4a indicate a reduction toward the infinite-layer $NdNiO_2$ phase, but with a modest decrease in crystallinity. Electron microscopy measurements on an equivalent sister sample, shown in Fig. 4b, reveal the presence of defects and phase boundaries, as expected.

The film was cut in half before reduction, and both as-grown and reduced samples were measured using ToF-SIMS, yielding the intensity depth profiles in Fig. 4c, d. As before, the $NiO^-$, $NiO_2^-$, and $TiO_2^-$ peaks are normalized to their maximum value while the $H^-$ and $OH^-$ are normalized to the steady-state values in the substrate. The $H^-$ and $OH^-$ signals are slightly higher in the $SrTiO_3$ substrate than in the as-grown $NdNiO_3$ film. Upon reduction, $H^-$ and $OH^-$ increase at the surface of the films, and the lineshape of this increase only partially matches that of various peaks including $C_2^-$ and $Ca^+$ (see Supplementary Note 3), indicating that they do not solely originate from surface adsorbates introduced during the reduction process. Near the substrate interface, which has previously been shown to be the highest-quality region of the film[20,33], the $H^-$ intensity remains lower than in the $SrTiO_3$ substrate. Thus, while some insignificant hydrogen content may be introduced during the reduction process, it seems to be limited near the surface of these uncapped films.

We next present our findings on non-superconducting infinite-layer $Nd_{0.8}Sr_{0.2}NiO_2$/$SrTiO_3$ films, which are appropriately doped to result in superconductivity but were reduced aggressively at high temperatures (600 °C compared to ~300 °C); this reduction is enough to significantly hydrogen-dope a similar perovskite material, $BaZrO_3$[38]. Furthermore, a common practice is to cap perovskite nickelate films with $SrTiO_3$ prior to reduction to provide balanced strain for structural stability of the film throughout its entire thickness[6,20]. We, therefore, compare samples with and without a $SrTiO_3$ capping layer grown in situ on the 10 nm $Nd_{0.8}Sr_{0.2}NiO_3$ before reduction.

XRD measurements, shown in Fig. 5a, reveal that the crystalline quality of both capped and uncapped films prior to reduction is lower, with broader, lower-intensity film peaks. Importantly, while the (002) $Nd_{0.8}Sr_{0.2}NiO_3$ peak is sharp, the expected perovskite (001) film peak is suppressed; further higher-resolution measurements, shown in Supplementary Fig. 12, resolve the presence of potential Ruddlesden–Popper phase. As before, the topotactic reduction process reduces the $c$-axis lattice parameter, but the peak intensity drops dramatically. Transmission electron micrographs of these samples, such as that shown in Fig. 5b, reveal the segregation of the film into multiple crystalline phases and amorphous-like regions. Thus, unlike the uniform superconducting samples, the aggressive reduction of these films is non-uniform and disordered, resulting in increased mosaicity and a loss of crystalline quality. This is corroborated by electronic transport, as discussed further in the Supplementary

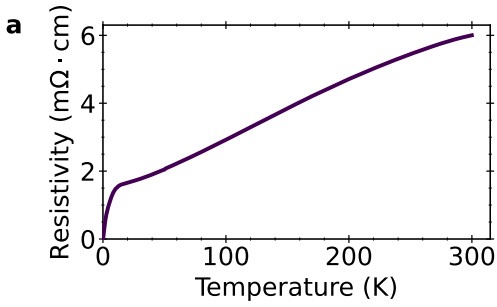

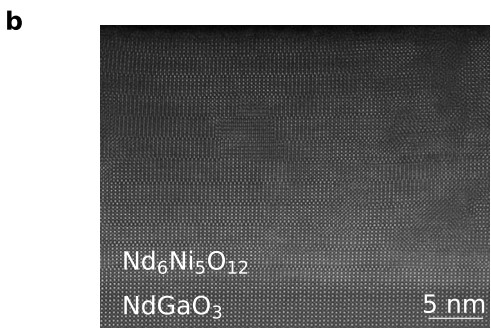

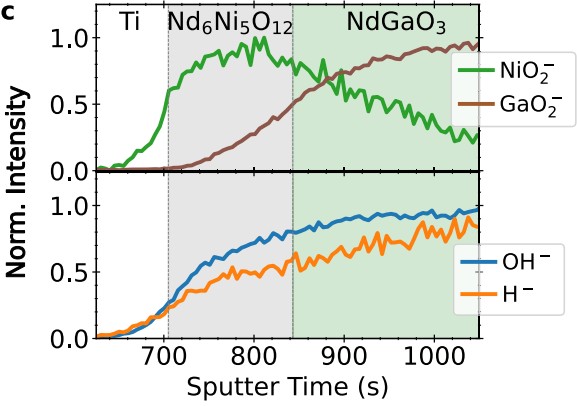

**Fig. 3 | Characterization of a superconducting Nd$_6$Ni$_5$O$_{12}$ film. a** Temperature-dependent resistivity of reduced Nd$_6$Ni$_5$O$_{12}$/NdGaO$_3$ showing a clear superconducting transition. The same data as in Ref. 10. **b** STEM image of the reduced, superconducting Nd$_6$Ni$_5$O$_{12}$ film and NdGaO$_3$ substrate. **c** SIMS depth profiles of superconducting Nd$_6$Ni$_5$O$_{12}$.

Information, which indicates that capped and uncapped films exhibit sharply different resistivities.

As shown in Fig. 5c, which plots SIMS measurements from the as-grown, uncapped Nd$_{0.8}$Sr$_{0.2}$NO$_3$ film, the initial transient region shows much higher yields of all ions which may indicate differences in crystallinity near the surface, potentially from the emergence of a polycrystalline scale layer in uncapped samples over time[21]. In the bulk region, the H$^-$ and OH$^-$ intensities are similar to the substrate.

Despite the significant difference in crystalline quality and reduction conditions, the effects of reduction are similar to our other observations. Figure 5d shows the integrated peak intensities in the reduced SrTiO$_3$/Nd$_{0.8}$Sr$_{0.2}$NiO$_2$ bilayer. Once again, H$^-$ and OH$^-$ intensities are elevated at the cap/film interface—though over a broader spatial extent—with almost a 50% increase over the baseline in the substrate. Thus, while the SrTiO$_3$ cap appears to trap hydrogen, this enhancement is again far below the orders of magnitude that would be expected for significant hydrogen incorporation, remaining within a factor of two of the substrate values.

## Theoretical Calculations

To further understand the lack of hydrogen incorporation in the nickelates analyzed above via SIMS (both superconducting and non-superconducting), density-functional theory (DFT)-based calculations were performed to explore the energetics of topotactic hydrogen in infinite-layer $R$NiO$_2$ ($R$ = rare-earth, both doped and undoped) as well as in the quintuple-layer nickelate Nd$_6$Ni$_5$O$_{12}$. To investigate whether it is energetically favorable to intercalate hydrogen, we compute the hydrogen binding energy ($E_b$) for the topotactic process as done in previous work[28]:

$$E_b = \{E[R\text{NiO}_2] + n \times \mu[H] - E[R\text{NiO}_2\text{H}]\}/n, \quad (1)$$

where $E[R\text{NiO}_2]$ and $E[R\text{NiO}_2\text{H}]$ are the total energies for the infinite-layer $R$NiO$_2$ and hydride-oxide $R$NiO$_2$H compounds, $\mu[H] = E[\text{H}_2]/2$ is the chemical potential of H, and $n$ represents the number of H atoms in the (super)cell. Analogous expressions are used for $R_{0.75}$(Sr,Ca)$_{0.25}$NiO$_2$ and Nd$_6$Ni$_5$O$_{12}$. A positive (negative) $E_b$ indicates that the topotatic hydrogen intercalation is favorable (unfavorable). The calculated binding energies are summarized in Fig. 6. We find that the incorporation of H into $R$NiO$_2$, $R_{0.75}$(Sr,Ca)$_{0.25}$NiO$_2$, Nd$_6$Ni$_5$O$_{12}$ is systematically unfavorable, in agreement with experiments (only for LaNiO$_2$ a very small positive E$_b$ value of approximately 10 meV/H is obtained).

## Discussion

In summary, we searched for hydrogen across a wide range of superconducting and non-superconducting layered nickelate films, with different cation and dopant chemistry, structures, growth methods, reduction conditions, and crystalline quality. Not only did we find no significant concentrations of hydrogen in superconducting films, but we were also unable to use excessive reduction temperature or time to force significant amounts of hydrogen into these structures. These results are consistent with first-principles calculations which show that hydrogen incorporation is energetically unfavorable in both infinite-layer and quintuple-layer nickelates. At most, we observed increased concentrations by a factor of two from the trace amounts already present within the substrates. Furthermore, hydrogen, as hydride or hydroxide ions (H$^-$ and OH$^-$), was more likely to be found in SrTiO$_3$ caps or in the substrates than in the nickelate films themselves. This propensity for hydrogen to appear in higher concentrations in SrTiO$_3$ capping layers and SrTiO$_3$/nickelate interfaces is interesting in the context of recent work showing the important role such capping layers can play in facilitating the reduction process[21].

It should be noted that our measurements generally reveal as-grown samples with hydrogen levels at or below the SIMS detection limit prior to reduction, although it is, of course, not possible to completely eliminate hydrogen from any material system. CaH$_2$ does appear to introduce hydrogen into the system, as evidenced by changes in both film and substrate levels in as-grown La$_{0.78}$Ca$_{0.22}$NiO$_3$ and reduced La$_{0.78}$Ca$_{0.22}$NiO$_2$ films, for example. However, topotactic reduction appears unable to introduce hydrogen into these nickelates at the levels near ANiO$_2$H$_{0.25}$ (A = La,Sr,Nd,Ca) previously cited as critical doping for superconductivity[27]. At such high levels of incorporated hydrogen, Ding et al. observed H$^-$ intensities approximately 40× to 60× the substrate concentration, and approaching a factor of 200× to 600× near the film surface. We find no evidence of such large relative H$^-$ intensities in the films studied in this work.

Therefore, although superconductivity is highly sensitive to reduction optimization, this is likely due to the crystalline quality and oxygen stoichiometry, and not hydrogen stoichiometry. Of course, this study does not demonstrate that superconductivity requires the complete absence of incorporated hydrogen. This work instead indicates that many films appear resistant to hydrogen infiltration and that superconductivity may be readily realized at very low hydrogen levels

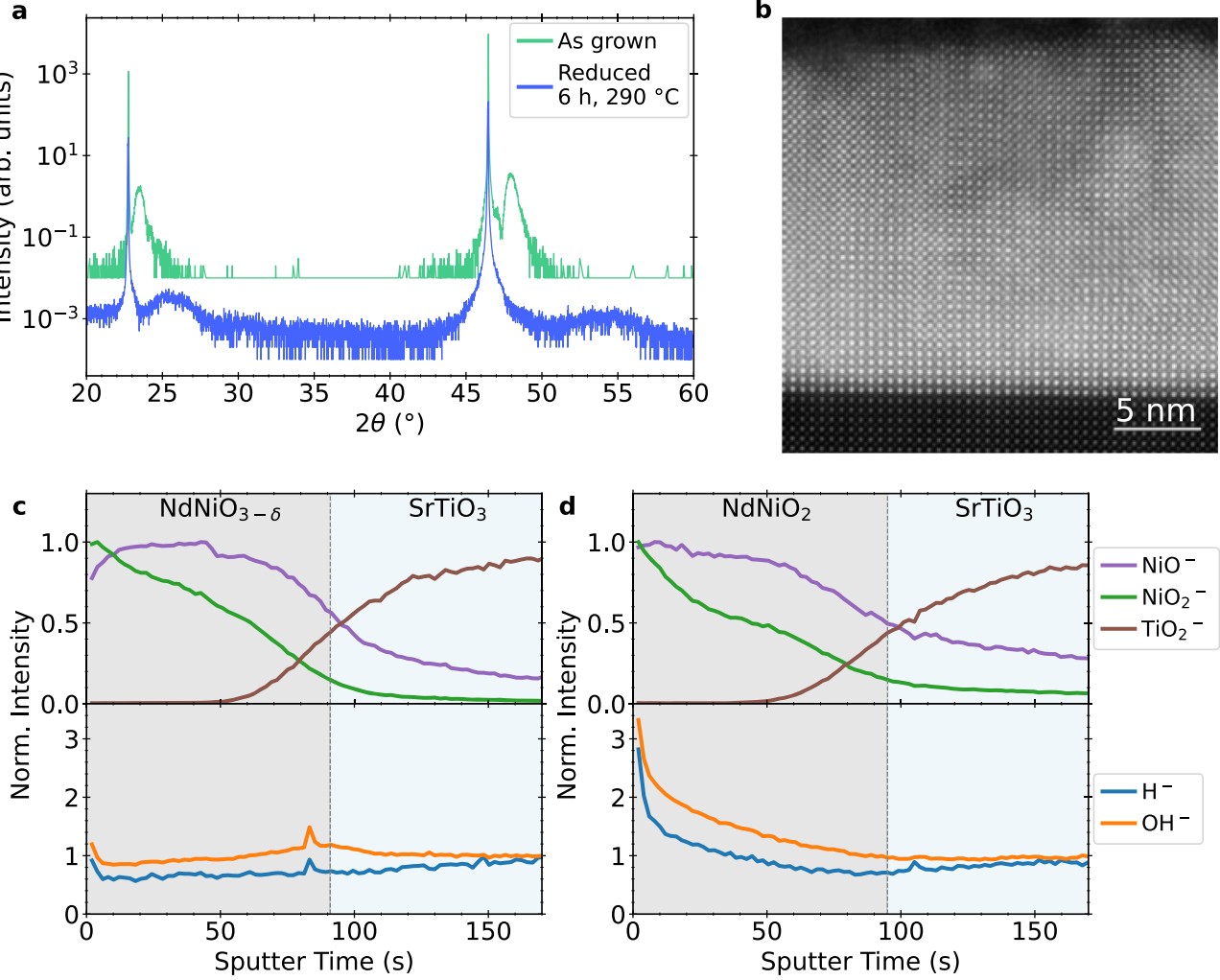

**Fig. 4 | Characterization of non-superconducting NdNiO₂ films. a** XRD of as-grown NdNiO₃ and reduced NdNiO₂ films showing partial reduction to the infinite-layer phase. **b** STEM of an equivalent sister NdNiO₂ sample revealing extended defects concentrated near the surface. **c** SIMS depth profile of the as-grown NdNiO₃

film. **d** SIMS depth profile of the NdNiO₂ film, reduced for an extended time of 6 h at 290 °C. SIMS shows increased hydrogen concentration at the surface even without full oxygen removal.

for which no theoretical evidence supports hydrogen-mediated superconductivity.

*Note added:* Recently, independent SIMS experiments performed by Zeng et al.[40] and Gonzalez et al.[41] also concluded that extensive hydrogen incorporation is unnecessary for superconductivity in the infinite-layer nickelates, in agreement with this work.

## Methods

### Sample synthesis: thin film deposition and reduction

**La₁₋ₓ(Ca,Sr)ₓNiO₂ films.** Thin films, ≈ 6 nm thick, of the infinite-layer nickelates La₀.₇₈Ca₀.₂₂NiO₂ and La₀.₈Sr₀.₂NiO₂ were grown on SrTiO₃ (001) substrates using pulsed laser deposition (PLD) and CaH₂ topotactic reduction[33,34]. SrTiO₃ (001) substrates were etched with hydrofluoric acid and annealed in air at 900 °C for 90 min before deposition. This is to maximize the TiO₂ termination which serves to minimize disordered Ruddlesden–Popper type growth. We first grow the perovskite phase using PLD with the following optimal set of parameters: $T_{growth}$ = 575 °C, $P_{O_2}$ = 150 mTorr (1 Torr = 133.322 Pa), $J$ = 2.5 J/cm². Afterwards, the film was post-annealed at growth temperature under the same oxygen partial pressure for 10 min followed by cooling at 8 °C/minute. The topotactic phase transition to the infinite-layer phase was mediated by the substrate strain and performed in the same PLD vacuum

chamber with a base pressure of less than 1 × 10⁻⁶ Torr. The reduction environment was achieved by heating approximately 0.1 g of CaH₂ powder to obtain a (H₂ and other species) pressure in the range of approximately 0.1–0.3 Torr. Samples are annealed at 340 °C for 1 h. After reduction, samples were capped with amorphous SrTiO₃ at room temperature using PLD to protect the surface from reoxidation. Mild oxidation damage to the top nickelate surface can be expected in this process.

**Nd₆Ni₅O₁₂ and NdNiO₂ Films.** We use ozone-assisted molecular beam epitaxy (MBE) to synthesize the precursor Nd₆Ni₅O₁₆/NdGaO₃ (110) and NdNiO₃/SrTiO₃ (001) films in Figs. 3 and 4, respectively. To calibrate the nickel and neodymium elemental fluxes, we synthesize NiO on MgO (001) and Nd₂O₃ on yttria-stabilized zirconia (YSZ (111)), then measure the film thickness via X-ray reflectivity. Next, we synthesize NdNiO₃/LaAlO₃ (001) and use the c-axis lattice constant and film thickness to refine the Nd/Ni ratio and monolayer dose, respectively. Using the optimized neodymium and nickel shutter times from the synthesis of NdNiO₃/LaAlO₃, we synthesize the Ruddlesden–Popper nickelates *via* monolayer shuttering. Both NdNiO₃ and Ruddlesden–Popper nickelates are synthesized at a substrate temperature of 500–600 °C with approximately 1.0 × 10⁻⁶ Torr distilled ozone (Heeg Vacuum Engineering). The MBE synthesis

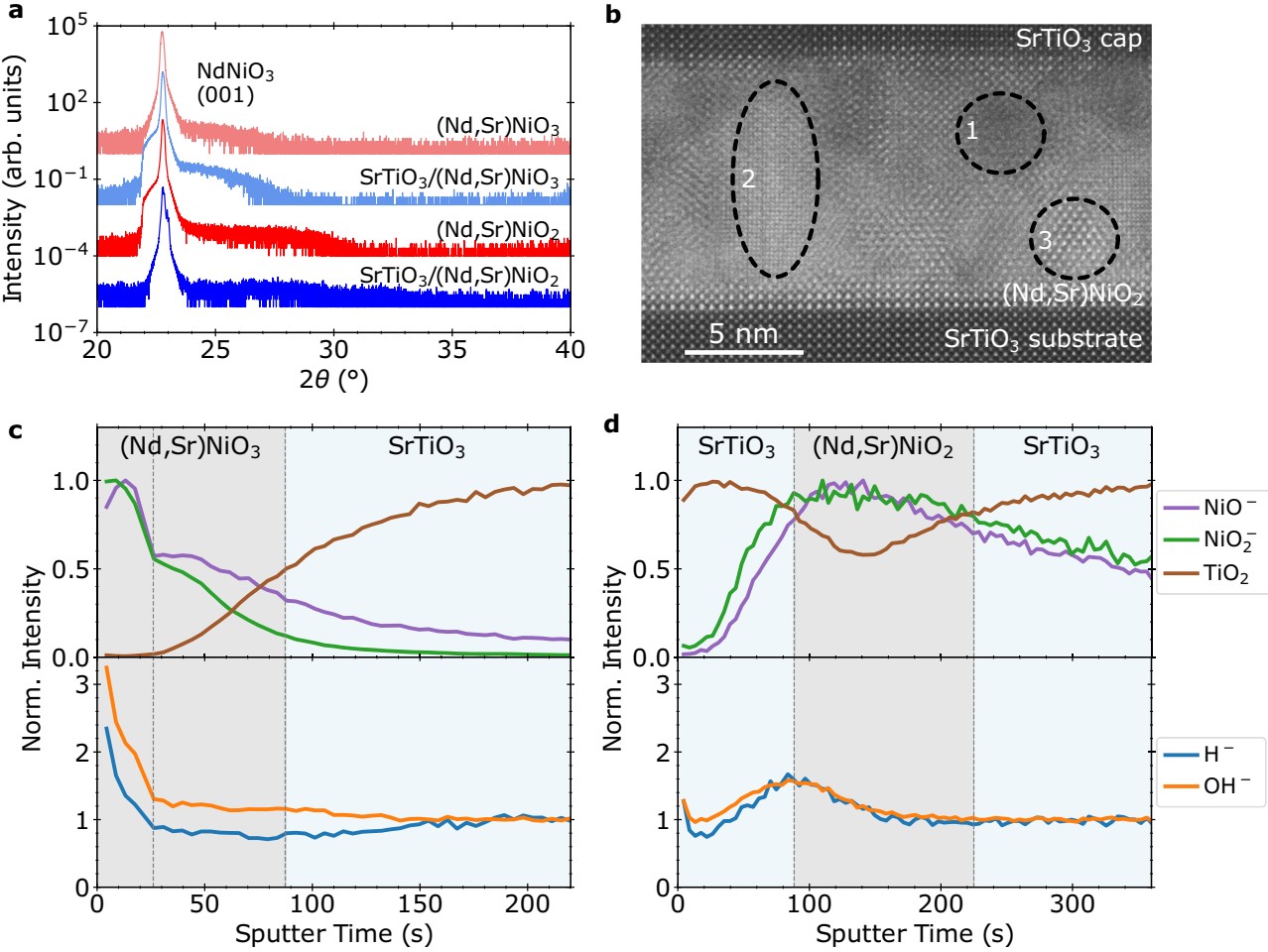

**Fig. 5 | Characterization of $Nd_xSr_{1-x}NiO_3$ and non-superconducting $Nd_xSr_{1-x}NiO_2$ films. a** XRD on $Nd_{0.8}Sr_{0.2}NiO_3$ films grown by PLD, with and without a 10 nm $SrTiO_3$ cap, showing a weak film (001) peak both before and after reduction, indicating a low-crystallinity as-grown film. Reduction further lowers crystallinity. **b** Atomic-resolution cross-sectional HAADF-STEM micrograph from the reduced $SrTiO_3/Nd_{0.8}Sr_{0.2}NiO_2$ film showing amorphous (marked as 1) and crystalline regions (marked as 2 and 3). The low-crystalline quality of the film after reduction is clearly visible and agrees with the XRD data. **c** SIMS depth profile of the as-grown $Nd_{0.8}Sr_{0.2}NiO_3$ film indicates a separate surface layer. **d** SIMS depth profile of the reduced $Nd_{0.8}Sr_{0.2}NiO_2$ film with a $SrTiO_3$ cap, showing non-negligible but small hydrogen incorporation at the $SrTiO_3$ cap/$Nd_{0.8}Sr_{0.2}NiO_2$ interface.

conditions and calibration scheme are described in Refs. 10,22; similar techniques were also used in Refs. 42,43.

The perovskite and Ruddlesden-Popper films are reduced to the square-planar phase via $CaH_2$ topotactic reduction. The following methods are similar to those used elsewhere[10,20,44]. First, the as-grown films are cut into identical pieces, and the pieces to be reduced are tightly wrapped in aluminum foil (All-Foils) to avoid direct contact between the film and $CaH_2$. Each film is then placed in a borosilicate tube (Chemglass Life Sciences) with approximately 0.1 g of $CaH_2$ pieces (>92%, Alfa Aesar). The borosilicate tube is pumped down to <0.5 mTorr, sealed, and then heated for several hours at 290 °C in a convection oven (Heratherm, Thermo Fisher Scientific) with a 10 °C min⁻¹ heating rate.

**$Nd_{0.8}Sr_{0.2}NiO_3$ films.** Polycrystalline targets of $NdNiO_3$ and $Nd_{0.8}Sr_{0.2}NiO_3$ were prepared by the liquid-mix technique[45,46]. A 10 nm thick $Nd_{0.8}Sr_{0.2}NiO_3$ films were grown on (001) $SrTiO_3$ substrates using a Neocera PLD system equipped with an in-situ RHEED (Staib Instruments, Germany). The depositions were conducted using a KrF excimer laser operating at 2 Hz with a fluence of 1.5 J cm⁻². During the deposition, a dynamic oxygen pressure of 150 mTorr was maintained, and the substrate temperature was 735 °C. The optional 10 nm thick $SrTiO_3$ capping layer was grown at the same condition as the film. After

the deposition, all samples were in-situ annealed at the deposition temperature in an oxygen atmosphere of 500 Torr for 30 min and subsequently cooled to room temperature at a rate of 15 °C min⁻¹.

The as-grown films were sealed in evacuated (approximately 1 mTorr) quartz ampoules with 0.1 g $CaH_2$ powder (90%–95%, Thermo Scientific Chemicals). The ampoules were then baked in a muffle furnace at 600 °C for up to 10 h. The temperature ramp rate was fixed at 10 °C min⁻¹. Once the ampoules were opened, the reduced films were immediately rinsed in *n*-butanol and isopropanol in an ultrasonic bath for 3 min.

**X-ray diffraction**

X-ray diffraction (XRD) measurements were performed at room temperature before and after reduction by each group on commercially available X-ray diffractometers using Cu K$\alpha_1$ ($\lambda = 1.5406$ Å) radiation.

**X-ray reflectometry**

X-ray reflectometry (XRR) measurements were performed at ambient conditions in a horizontal configuration using a Rigaku SmartLab diffractometer. The incident beam was collimated using the parallel beam slit and an incident slit of 30 μm height to improve *Q*-resolution. The Cu K$\alpha_1$ wavelength ($\lambda = 1.5406$ Å) was isolated by using a Ge-(220) × 2 monochromator. The scattered beam was further collimated by a

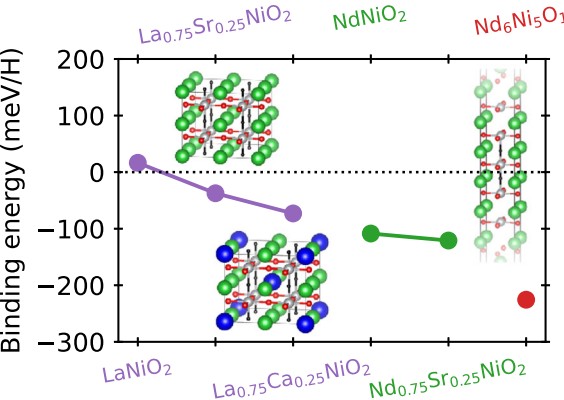

**Fig. 6 | Topotactic-H energetics for superconducting nickelates.** Binding energies for hydrogen in superconducting layered nickelates, where a positive (negative) binding energy indicates favoring (disfavoring) H-intercalation. Crystal structures indicate the positions of topotatic-H used for (top) (La,Nd)NiO$_2$, (bottom) (La,Nd)$_{0.75}$(Sr,Ca)$_{0.25}$NiO$_2$, and (right) Nd$_6$Ni$_5$O$_{12}$. Solid lines are guides to the eye connecting related materials with the same parent phase e.g., LaNiO$_2$ and the associated doped compounds.

0.2 mm receiving slit. The data were reduced using the *reductus* web-service[47] and fit to a slab model using Refl1D[48].

**Time-of-flight secondary ion mass spectroscopy**
Time-of-flight SIMS was performed using an IONTOF IV (Münster, Germany) equipped with a 20 keV Ar$_{2600\pm1000}^+$ cluster source for sputtering, a 30 keV Bi$_3^+$ liquid metal ion source for analysis, and a time-of-flight mass analyzer. Depth profiling was performed in non-interlaced mode with 1 scan of analysis with a lateral resolution of (128 × 128) pixels, 10 scans of sputtering, and at least 0.5 s of charge compensation per cycle, where both the analysis and sputter rasters were kept inside a (500 × 500) μm area. The corresponding ion doses were 1.9 × 10$^9$ ions/cm$^2$ (0.12 pA) for Bi$_3^+$, and between 2.1 × 10$^{14}$ ions/cm$^2$ to 2.6 × 10$^{14}$ ions/cm$^2$ (5.1–6.4 nA) per cycle for the cluster source due to day-to-day fluctuations in the beam current. On especially insulating samples or substrates, a small drop of silver paint was used to electrically contact the sample surface to the sample holder for further charge compensation.

For reliable detection of H$^-$ ions, contributions from residual gases were minimized by keeping the chamber pressure below 5 × 10$^{-7}$ Pa. Both negative and positive ions were analyzed at separate spots, and the signal rastered over multiple spots was averaged after normalizing for the highest intensity ion unique to the substrate (TiO$_2^-$ or GaO$_2^-$).

Spectra were analyzed using SurfaceLab to define a region of interest, perform mass calibrations, identify peaks with the appropriate compounds, and extract the total integrated peak intensity as a function of sputter time. As many molecular compounds can have similar mass, peak assignments were made carefully, considering factors such as mass offset, isotopic distribution, and similarity in profile shape to other known oxide and hydroxide species.

**Electron microscopy**
All cross-sectional STEM specimens were prepared by the standard focused ion beam (FIB) lift-out procedure and imaged in high-angle annular dark-field (HAADF)-STEM configuration. The instrument, processing, and experimental details for specific samples are as follows:
- Nd$_6$Ni$_5$O$_{12}$. Preparation: Thermo Fisher Scientific Helios G4 UX and FEI Helios 660 FIBs. Imaging: probe-corrected Thermo Fisher Scientific Spectra 300 X-CFEG operating at 300 kV, 19 mrad convergence semi-angle, 33 mrad inner collection angles.

- NdNiO$_2$. Preparation: FEI Helios 660 FIBs with final polishing at 5 kV accelerating voltage and 41 pA probe current. Imaging: Thermo Fisher 615 Scientific Titan Themis Z G3 operating at 200 kV, 18.9 mrad convergence semi-angle, and 68 (280) mrad inner (outer) collection angles.
- SrTiO$_3$-capped (Nd,Sr)NiO$_{3-x}$ and (Nd,Sr)NiO$_2$. Preparation: Thermo Fisher Scientific Helios G4 UXe PFIB Dual Beam, with final polishing at 5 kV accelerating voltage. Imaging: Thermo Fisher Scientific Spectra 200 operating at 200 kV, 25 mrad convergence semi-angle, 54 (200) mrad inner (outer) collection angles, and probe current of approximately 20 pA. Energy dispersive X-ray spectroscopy (EDS) chemical mapping was performed on the same instrument with Bruker Dual-X X-ray detectors an electron beam current of approximately 100 pA.

**Computational methods**
Density-functional theory (DFT)-based calculations were performed to theoretically explore the energetics of topotactic hydrogen in the infinite-layer nickelate $R$NiO$_2$ ($R$ = La, Nd, both doped and undoped) as well as in the quintuple-layer nickelate Nd$_6$Ni$_5$O$_{12}$. For $R$NiO$_2$H$_\delta$, $R_{0.75}$(Sr,Ca)$_{0.25}$NiO$_2$H$_\delta$, and Nd$_6$Ni$_5$O$_{12}$H$_\delta$ ($\delta$ = 0, 1) structural relaxations were performed using the VASP code[49–51] with the the Perdew–Burke–Ernzerhof version of the generalized gradient approximation (GGA-PBE)[52]. For the infinite-layer materials ($R$NiO$_2$ and $R_{0.75}$(Sr,Ca)$_{0.25}$NiO$_2$) up to a 2 × 2 × 2 supercell was used to accommodate the appropriate H content and/or (Sr,Ca)-doping level. We place the topotatic-H at the positions of the (removed) apical oxygens as this was shown to be the most energetically favorable position for H-incorporation from previous works[27–30]. GGA-PBE was chosen as it provides lattice constants in close agreement with experimental data, as shown in Supplementary Note H. A $\Gamma$-centered 13 × 13 × 15 (9 × 9 × 11) $k$-mesh was used for the 1 × 1 × 1 unit cells (2 × 2 × 2 supercells) with a 0.05 eV Gaussian smearing. For Nd$_6$Ni$_5$O$_{12}$, a $\Gamma$-centered 9 × 9 × 9 $k$-mesh with a 0.05 eV Gaussian smearing was used. The size of the plane-wave basis sets was set with a kinetic energy cut-off of 520 eV. For $R$ = Nd, we have used a pseudopotential where the Nd(4$f$) electrons are frozen in the core. To compute the chemical potential of hydrogen ($\mu$[H]), we optimized an H$_2$ dimer in 15$^3$ Å$^3$ box with energy cutoff set to 325 eV.

**Data availability**
The data supporting this study have been deposited in Figshare.

**Code availability**
Analysis was performed by open-source Python packages, including NumPy, matplotlib, and SciPy. SIMS data reduction was performed using the commercial software SurfaceLab 7. All density-functional theory calculations were performed with the Vienna Ab-Initio Simulation Package (VASP).

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

## Acknowledgements

D.F.S., G.A.P., and J.A.M. acknowledge support from the US Department of Energy, Office of Basic Energy Sciences, Division of Materials Sciences and Engineering, under Award No. DE-SC0021925. L.E.C. and A.A. acknowledge support from the Ministry of Education (MOE), Singapore, under its Tier-2 Academic Research Fund (AcRF), Grants No. MOET2EP50121-0018 and MOE-T2EP50123-0013, and the SUSTech-NUS Joint Research Program. J.R. and M.S. acknowledge support from an ARO MURI program with award no. W911NF-21-1-0327, and the National Science Foundation of the United States under grant number DMR-2122071. R.K.P. and S. Middey acknowledge MHRD, Government of India for financial support under the STARS research funding scheme (grant number: STARS/APR2019/PS/156/FS). J.A.M., A.S.B., and H.L. acknowledge support from NSF grant no. DMR-2323971. I.E. and Y.Z. were supported by the Rowland Institute at Harvard. K.J. and Y.T.S. acknowledge support from USC Viterbi startup funding and the USC Research and Innovation Instrumentation Award. B.H.G. and L.F.K acknowledge support from the Platform for the Accelerated Realization, Analysis, and Discovery of Interface Materials (PARADIM) and the Packard Foundation. Materials growth and electron microscopy were supported in part by PARADIM under NSF Cooperative Agreement no. DMR-2039380. Focused ion beam sample preparation was performed in part at the Harvard University Center for Nanoscale Systems (CNS); a member of the National Nanotechnology Coordinated Infrastructure Network (NNCI), which is supported by the National Science Foundation under NSF award no. ECCS-2025158. Transmission electron microscopy was carried out in part through the use of MIT.nano's facilities. Electron microscopy data were acquired in part at the Core Center of Excellence in Nano Imaging at USC. Electron microscopy was performed in part at the Cornell Center for Materials Research (CCMR) Shared Facilities, which are supported by the NSF MRSEC Program (No. DMR-1719875). P.P.B. and P.Q. received funding from the NRC RAP. G.A.P. acknowledges additional support from the Paul and Daisy Soros Fellowship for New Americans. D.F.S. and G.A.P acknowledge support from the NSF Graduate Research Fellowship Grant DGE-1745303. We thank Kyuho Lee for insightful discussions. We also thank Kerry Sieben for X-ray assistance. We thank Hanjong Paik for supporting the growth of the $n = 5$ superconducting sample. Research was performed in part at the NIST Center for Nanoscale Science and Technology. Certain commercial equipment, instruments, software, or materials are identified in this paper in order to specify the experimental procedure adequately. Such identifications are not intended to imply recommendation or endorsement by NIST, nor it is intended to imply that the materials or equipment identified are necessarily the best available for the purpose.

## Author contributions

P.P.B., S. Muramoto, and A.J.G. performed and analyzed ToF-SIMS measurements. P.P.B., A.J.G., and P.Q. performed XRR and analyzed and interpreted all reflectometry data. L.E.C. deposited and reduced La$_{0.8}$Sr$_{0.2}$NiO$_3$ and La$_{0.78}$Ca$_{0.22}$NiO$_3$ films, and performed electronic characterization. NdNiO$_3$ and Nd$_6$Ni$_5$O$_{16}$ films were fabricated by G.A.P., D.F.S., and Q.S., while G.A.P. and D.F.S. performed reduction and electronic characterization. Nd$_{0.8}$Sr$_{0.2}$NiO$_3$ films were deposited and initially characterized by R.K.P. and S. Middey, and reduced by M.S. TEM measurements were performed and analyzed by Y.Z., I.E., K.J., Y.T.S., B.H.G., and L.F.K. H.L. and A.S.B. performed the theoretical calculations. The study was designed by A.J.G., P.P.B., P.Q., D.F.S., J.A.M., A.A., and J.R. The paper was written by P.P.B. and A.J.G. with input from all authors.

## Competing interests

The authors declare no competing interests.

## Additional information

[1]NIST Center for Neutron Research, National Institute of Standards and Technology, Gaithersburg, MD 20899, USA. [2]Department of Physics, Harvard University, Cambridge, MA 02138, USA. [3]Department of Physics, Faculty of Science, National University of Singapore, Singapore 117551, Singapore. [4]Material Measurement Laboratory, National Institute of Standards and Technology, Gaithersburg, MD 20899, USA. [5]Mork Family Department of Chemical Engineering and Materials Science, University of Southern California, Los Angeles, CA 90089, USA. [6]Core Center for Excellence in Nano Imaging, University of Southern California, Los Angeles, CA 90089, USA. [7]Department of Physics, Indian Institute of Science, Bengaluru 560012, India. [8]Department of Physics, Arizona State University, Tempe, AZ 85287, USA. [9]The Rowland Institute at Harvard, Harvard University, Cambridge, MA 02138, USA. [10]Core Center for Excellence in Nano Imaging, University of Southern California, 925 Bloom Walk, Los Angeles, CA 90089, USA. [11]School of Applied and Engineering Physics, Cornell University, Ithaca, NY 14853, USA. [12]Kavli Institute at Cornell for Nanoscale Science, Ithaca, NY 14853, USA. [13]Max Planck Institute for Chemical Physics of Solids, 01187 Dresden, Germany. [14]Ming Hsieh Department of Electrical and Computer Engineering, University of Southern California, Los Angeles, CA 90089, USA. [15]These authors contributed equally: Purnima P. Balakrishnan, Dan Ferenc Segedin, Lin Er Chow ✉e-mail: purnima.balakrishnan@nist.gov; j.ravichandran@usc.edu; ariando@nus.edu.edu; mundy@fas.harvard.edu; alexander.grutter@nist.gov

