## [Peer Review File · Nature Communications]

REVIEWER COMMENTS

Reviewer #1 (Remarks to the Author):

The manuscript by Purnima P. Balakrishnan et al. presents a comprehensive study using secondary ion mass spectrometry (SIMS) to investigate hydrogen distribution across a variety of superconducting and non-superconducting layered nickelate films. The authors report a negligible concentration of hydrogen in the superconducting films, challenging previous findings in the field. Furthermore, their experiments did not detect significant hydrogen incorporation under conditions of extended reduction time or increased temperature. From these observations, the authors conclude that hydrogen is not essential for superconductivity in topotactically reduced nickelates.

While the findings are technically sound and might hold interest for researchers in the field of superconducting nickelates, the broader scientific insights provided by the study appear limited. The scope and impact of the findings do not align with the high standards expected by Nature Communications. Additionally, the presentation and discussion of the SIMS data could benefit from further clarification to enhance understanding and reproducibility.

Given these considerations, I recommend that the manuscript may be better suited for publication in a specialized journal that focuses on the niche area of superconducting materials. This would allow the work to reach an audience well-versed in the specific techniques and topics discussed, potentially fostering more detailed technical discussions and follow-up studies.

Some suggestions are as follows:

Figure 2 contrasts the hydrogen concentration in different superconducting films and concludes that hydrogen doping is not essential for superconductivity in infinite layer nickelates. However, the presence of hydrogen, albeit at a low concentration, in the superconducting films calls into question the robustness of this conclusion. Can the authors clarify the rationale behind this assertion?

The manuscript notes that the infinite layer structure phase is not uniformly distributed across the reduced films, despite their superconductivity. Could the authors discuss how defects or secondary phases might influence the accuracy of hydrogen measurements in these superconducting samples?

The authors state that Secondary Ion Mass Spectrometry (SIMS) provides "a nanometer-resolved understanding of the chemical composition with depth." Given that the sample thickness is only a few nanometers (e.g., 6 nm as shown in Figure 2), the depth resolution of SIMS becomes critically important. It would be beneficial for the authors to provide specific figures for the depth resolution achieved in these measurements.

In Figures 2e and 2f, there is a pronounced signal for NiO₂ in the substrate and TiO₂ in the film. Could the authors explain the origins of these strong signals and how the interfaces in the intensity profiles were determined? Additionally, the superconducting La_{0.78}Ca_{0.22}NiO₂ and La_{0.8}Sr_{0.2}NiO₂ samples both have a thickness of 6 nm, yet the sputter times differ significantly (approximately 100s for LCNO and 300s for LSNO). What accounts for this discrepancy?

When comparing results in the manuscript with those reported elsewhere, it is essential to provide references to support these comparisons, as noted around Line 197 on Page 3. Could the authors include the necessary citations to strengthen their discussion?

Reviewer #2 (Remarks to the Author):

My field of expertise lies in TOF-SIMS. I found this work clearly showed that hydrogen incorporation was not required for superconductivity in these Nickelates. The authors were detailed in their explanations of how the SIMS data was collected and interpreted, and I see no need for revisions. I especially enjoyed seeing the attention to detail presented in supplementary sections on the SIMS data analysis. Also the details covered in supplementary section E are often not clearly stated in publications, even though I feel they should be. I congratulate the authors on their attention to detail in interpreting their SIMS data.

Reviewer #3 (Remarks to the Author):

The present manuscript by Balakrishnan et al. aims to assess the level of importance of possible hydrogen intercalation into the lattice of square planar nickelates in relation to their superconducting properties. The tools used here are secondary ion mass spectrometry complemented with density functional theory calculations. Acquiring accurate and reliable information about the role of topotactic hydrogen in these novel and highly debated systems would greatly advance our microscopic understanding of their superconductivity.

So far, however, hydrogen in the nickelates has remained a rather controversial issue. On one hand, several theoretical studies (Refs. 24-26 of the manuscript) consistently find it to be energetically unfavourable for hydrogen to enter the lattice of chemically hole doped nickelates. On the other hand, in a recent experimental work by Ding et al. (Ref. 23) superconductivity was observed only in samples with high measured concentrations of hydrogen.

In spite of using an identical experimental technique, the conclusions of the present work are radically different: the authors argue not only that hydrogen is irrelevant for the nickelate superconductivity but also that its concentration is always negligibly small in both superconducting and non-superconducting samples, irrespective of sample quality. These experimental findings seem to be in agreement with DFT calculations showing that hydrogen intercalation is unfavourable for the considered nickelate systems except LaNiO_2 .

As far as the theoretical part is concerned, I find the presented results scientifically sound within the capabilities of the adopted DFT approximations, which are clearly documented. The occasional differences with previously reported results for binding energies, in particular that of NdNiO_2 , are attributed to using an improved DFT functional, but overall the qualitative trends regarding hydrogen intercalation are well reproduced.

What I find concerning, however, is the dramatic difference between the experimental data in the present work and in Ref. 23, both using same technique for measuring hydrogen content. If this technique allows for such substantial fluctuations in measuring the hydrogen content, then its reliability and scientific value are seriously questionable. The authors have to provide an explanation as to why their data should be considered more reliable than the work of Ding et al.

Reviewer #4 (Remarks to the Author):

I co-reviewed this manuscript with one of the reviewers who provided the listed reports.

REVIEWER COMMENTS

Reviewer #1 (Remarks to the Author):

The manuscript by Purnima P. Balakrishnan et al. presents a comprehensive study using secondary ion mass spectrometry (SIMS) to investigate hydrogen distribution across a variety of superconducting and non-superconducting layered nickelate films. The authors report a negligible concentration of hydrogen in the superconducting films, challenging previous findings in the field. Furthermore, their experiments did not detect significant hydrogen incorporation under conditions of extended reduction time or increased temperature. From these observations, the authors conclude that hydrogen is not essential for superconductivity in topotactically reduced nickelates.

While the findings are technically sound and might hold interest for researchers in the field of superconducting nickelates,

Response:

We appreciate the referee's assessment of our work as being technically sound, with conclusions that are well-supported by the data.

the broader scientific insights provided by the study appear limited. The scope and impact of the findings do not align with the high standards expected by Nature Communications. Additionally, the presentation and discussion of the SIMS data could benefit from further clarification to enhance understanding and reproducibility.

Given these considerations, I recommend that the manuscript may be better suited for publication in a specialized journal that focuses on the niche area of superconducting materials. This would allow the work to reach an audience well-versed in the specific techniques and topics discussed, potentially fostering more detailed technical discussions and follow-up studies.

Response:

We appreciate the opportunity to clarify the importance of our work in a broader context. High-temperature cuprate superconductivity (HTS) in the cuprates remains a defining problem in condensed-matter physics. One approach to address this problem has been the study of alternative transition metal oxides with similar structures and 3d electron count that are suggested as proxies for cuprate physics. Nickelates were prime targets in this context from the beginning (given the proximity of Ni to Cu in the periodic table) but it took over 30 years for the first superconducting nickelates to be realized: hole-doped infinite-layer nickelates $RNiO_2$ R=rare-earth (*Nature* **572**, 624 (2019)). Subsequently, superconductivity in the five-layer member of the same structural family $Nd_6Ni_5O_{12}$ was realized as well (*Nature Materials* **21**, 160 (2022)).

As the reduced nickelates contain Ni ions in an oxidation state at or close to 1+ (unstable for Ni), one can only get to these phases via oxygen reduction from parent compounds wherein Ni is in a more stable oxidation state. The prime oxygen reducing agent for these nickelates as of now is hydrogen.

Hence, understanding the role (or lack of it) of hydrogen incorporation for superconductivity in layered nickelates is crucial. Unfortunately, prior to our work, as clearly recognized by Referee 3 “Hydrogen in the nickelates has remained a rather controversial issue”, which he/she highlights in the context of contradictory findings between theory (*Phys. Rev. Lett.* 124, 166402 (2020)) and recent experiments (*Nature* **615**, 50 (2023)). With our work and our main finding, namely, that hydrogen is not necessary for superconductivity in reduced nickelates, **we lay to rest a key question which has remained a source of intense controversy since the discovery of superconductivity in infinite-layer nickelates.** This is crucial not only from a materials synthesis/sample preparation perspective restricted to nickelates but in the broader context of understanding the similarities and differences between layered nickelates and the cuprates for HTS.

We would also like to bring the referee’s attention to a recent editorial in *Nature Materials* (“*A deeper understanding*,” *Nature Materials* **23** 441 (2024)) which highlights the importance of reproducibility and careful examination of new superconducting families – especially the superconducting nickelates. The manuscript begins with “*The scientific community has become more aware of issues surrounding reproducibility. We should remain cognizant that as we develop better understanding we can revise our knowledge of a problem and scientific findings that are considered valid today might not be in the future.*” and ends with “*We look forward to seeing further discoveries in the phase diagrams of superconducting materials, with the ultimate hope of understanding the key ingredients for high-temperature superconductivity and making this field truly hot.*”

We agree strongly with these sentiments, and our study ideally meets this need. In the name of reproducibility, our study has brought together samples from multiple leading groups in the field for a combined study for the first time. For all the above reasons, we believe that our study is impactful enough to appeal to the broad audience of *Nature Communications*.

Some suggestions are as follows:

Figure 2 contrasts the hydrogen concentration in different superconducting films and concludes that hydrogen doping is not essential for superconductivity in infinite layer nickelates. However, the presence of hydrogen, albeit at a low concentration, in the superconducting films calls into question the robustness of this conclusion. Can the authors clarify the rationale behind this assertion?

Response:

We appreciate the opportunity to further clarify our conclusions. A prior study, ref. 23, claimed that a specific range of hydrogen doping around a very high value of $x = 0.25$ was required for superconductivity. We conclusively show in our work that this is not the case by giving multiple examples of both superconducting and non-superconducting samples with hydrogen content very different from that reported in ref. 23.

We do not claim a complete absence of hydrogen in our superconducting films, nor do we argue that trace hydrogen in the samples renders it impossible to stabilize superconductivity. Indeed, it is impossible to eliminate every single H-atom from a sample. **Rather, we find that in our samples, the hydrogen concentration is so low that it is not expected to determine whether superconductivity is observed.**

Although we do not agree with the quantification used in Ref. 23, we may use it to establish an approximate upper limit to the hydrogen content in our samples. Here we note that, based on their methodology, the highest H concentration, x , in the SrTiO_3H_x substrate was $x = 0.01$ for the $t = 420$ min sample. The average H^- intensity of the superconducting $(\text{La,Sr})\text{NiO}_2$ sample shown in Fig. 2 of the main text (reproduced as Figure R1 below) is less than 20% that of the substrate. So, assuming the highest possible concentrations in the substrate based on the methodology of Ref. 23, we estimate a worst-case H concentration of $\text{La}_{0.8}\text{Sr}_{0.2}\text{NiO}_2\text{H}_{0.002}$. The same calculation for the $(\text{La,Ca})\text{NiO}_2$ sample shown in Fig. 2 results in a worst-case H concentration of $\text{La}_{0.78}\text{Ca}_{0.22}\text{NiO}_2\text{H}_{0.016}$. The true levels are likely much lower than this limiting value.

Figure R1. ToF-SIMS depth profiles for the superconducting $\text{La}_{0.8}\text{Sr}_{0.2}\text{NiO}_2$ film shown in Fig. 2.

Similarly, we reproduce the non-superconducting NdNiO_2 film from Fig. 4 of the main text below as Figure R2, where the majority of the NdNiO_2 film contains H^- within a factor of two of the substrate. The same method then results in a “worst-case” stoichiometry of $\text{NdNiO}_2\text{H}_{0.013}$.

Figure R2. ToF-SIMS depth profiles for the NdNiO_2 films shown in Fig. 3.

Thus, both superconducting and non-superconducting films have, according to the methodology of Ref. 23, upper H concentration limits between 0.05 at% and 0.4 at%. These values are far outside the superconducting dome Ref. 23 reported. Though theoretical works disagree on the impact of high concentrations of hydrogen (e.g. Phys. Rev. B 108, 155147 (2023) and Phys. Rev. B 108, 174512 (2023)),

no extant theoretical studies in the literature indicate that such small concentrations have any bearing on the superconductivity.

To clarify these points in the manuscript, we altered the discussion to read:

“It should be noted that our measurements generally reveal as-grown samples with hydrogen levels at or below the SIMS detection limit prior to reduction, although it is of course not possible to completely eliminate hydrogen from any material system”

The previous version of this sentence used the phrasing “devoid of hydrogen” which did not precisely convey our intended meaning. We thank the referee for bringing this issue to our attention. Further, we note that the concluding paragraph has been revised to read:

“Of course, this study does not demonstrate that superconductivity requires the absence of incorporated hydrogen. The results of Ding et al. clearly demonstrate that it is possible to realize the superconducting state in samples which contain significant hydrogen concentrations. This work instead indicates that many films appear resistant to hydrogen infiltration, and that superconductivity may be readily realized at very low hydrogen levels for which no theoretical evidence supports hydrogen-mediated superconductivity.”

The manuscript notes that the infinite layer structure phase is not uniformly distributed across the reduced films, despite their superconductivity.

Response:

We appreciate the opportunity to further clarify our work. While reduction and strain effects can lead to different crystalline phases, all superconducting films measured in this study were uniform. In particular, the superconducting infinite-layer films were limited in thickness to ensure coherent strain across the entire depth of the sample, while TEM of the $n=5$ superconductor also shows a uniform square-planar phase. Our discussions on non-uniformity refer to the non-superconducting sample sets.

For clarity, we have revised the relevant section of the manuscript to now read:

“Thus, unlike the uniform superconducting samples, the aggressive reduction of these films is non-uniform and disordered rather than controlled, resulting in increased mosaicity and a loss of crystalline quality.”

Could the authors discuss how defects or secondary phases might influence the accuracy of hydrogen measurements in these superconducting samples?

Response:

We emphasize that we do not claim precise quantification of the hydrogen concentration anywhere in our manuscript, as an appropriate calibration for the relative sensitivity factor (RSF) has not been identified. However, it is important to discuss whether defects or secondary phases might influence the ability to compare measurements and infer trends.

There are two ways in which defects or secondary phases could be non-uniformly distributed across either:

- A. Vertical variation through the film, either in stoichiometry or in crystal structure.
- B. Lateral variation through the film due to phase segregation.

Neither are significantly present in the superconducting samples studied in our work; however, we can discuss the effects of these defects on SIMS measurements of our two sets of non-superconducting samples, undoped NdNiO_2 and aggressively-reduced $\text{Nd}_{0.8}\text{Sr}_{0.2}\text{NiO}_2$.

In the undoped NdNiO_2 shown in Fig. 4, reproduced here as Figure R3, partial reduction can result in vertical inhomogeneity, seen as extended defects in the top half of this uncapped film. In this case, the crystalline structure is not different enough to significantly change the SIMS intensity of any of the ion species, so we may approximate a constant sensitivity to different ions within the film.

Figure R3. A. XRD of as-grown and NdNiO_3 and reduced NdNiO_2 films showing partial reduction to the infinite-layer phase. B. STEM of an equivalent sister NdNiO_2 sample revealing extended defects concentrated near the surface. C. SIMS depth profile of the as-grown film NdNiO_3 film. D. SIMS depth profile of the NdNiO_2 film, reduced for an extended time of 6 h at 290 °C. SIMS shows increased hydrogen concentration at the surface even without full oxygen removal.

Figure R4. a. XRD on $\text{Nd}_{0.8}\text{Sr}_{0.2}\text{NiO}_3$ films grown by PLD, with and without a 10 nm SrTiO_3 cap, showing a weak film (001) peak both before and after reduction, indicating a low-crystallinity as-grown film. Reduction further lowers crystallinity. b. Atomic-resolution cross-sectional HAADF-STEM micrograph from the reduced $\text{SrTiO}_3/ \text{Nd}_{0.8}\text{Sr}_{0.2}\text{NiO}_2$ film showing amorphous (marked as 1) and crystalline regions (marked as 2 and 3). The low-crystalline quality of the film after reduction is clearly visible and agrees with the XRD data. c. SIMS depth profile of the as-grown $\text{Nd}_{0.8}\text{Sr}_{0.2}\text{NiO}_3$ film indicates a separate surface layer. d. SIMS depth profile of the reduced $\text{Nd}_{0.8}\text{Sr}_{0.2}\text{NiO}_2$ film with a SrTiO_3 cap, showing non-negligible but small hydrogen incorporation at the SrTiO_3 cap / $\text{Nd}_{0.8}\text{Sr}_{0.2}\text{NiO}_2$ interface.

Vertical inhomogeneity also exists to a greater degree in our uncapped as-grown $(\text{Nd,Sr})\text{NiO}_3$ films, seen here in Figure R4, reproduced from Fig. 5. This type of secondary phase may be caused by sample aging (i.e. surface oxidation) after exposure to atmosphere (Ref. 18 now at APL Mater. 12, 031132 (2024)), or strain-relieving extended defects. In this case, we do see a sharp and significant enhancement in all ion intensities, including H^- , in this region of the film (panel c). Since H^- increases similarly in intensity to other ions (NiO^- , NiO_2^-), we suspect a change in sputtering yield due to higher defect concentrations in this region. However, since the RSF may also change in this layer, we prefer not to make precise quantitative statements about the hydrogen content.

Lateral inhomogeneity can be seen as phase segregation after the aggressive reduction of the capped non-superconducting $(\text{Nd,Sr})\text{NiO}_2$ films, resulting in ~ 5 -10 nm regions with different crystalline

and amorphous structures (panel b). In contrast, our SIMS measurements spanned several hundred microns laterally, with submicron spatial resolution. Thus, any small-scale nonuniformities would be averaged over in these non-superconducting samples. SIMS then will provide an accurate laterally averaged picture of the depth-dependent hydrogen content of these films (panel d).

Thus, the reported inhomogeneities impact SIMS from only one sample subset, and do not alter the interpretation of SIMS data on these samples beyond confirming our impressions from STEM imaging. We emphasize that the inhomogeneities discussed do not extend to superconducting samples or impact the conclusions of the work. That being said, further work would be needed in this context, specifically studying the origin and types of defects and secondary phases which occur due to differences in reduction processing of these nickelate films, and whether those phases are more likely to absorb hydrogen. While a very interesting topic, that study is outside the scope of the present work.

The authors state that Secondary Ion Mass Spectrometry (SIMS) provides "a nanometer-resolved understanding of the chemical composition with depth." Given that the sample thickness is only a few nanometers (e.g., 6 nm as shown in Figure 2), the depth resolution of SIMS becomes critically important. It would be beneficial for the authors to provide specific figures for the depth resolution achieved in these measurements.

Response:

Unfortunately, our use of the word "resolution" here was somewhat imprecise and **we have modified the wording of this section to better reflect our meaning**. The most important issue is "layer separability" i.e. our ability to probe each individual layer. As a test case, we examine the data from uncapped as-grown (La,Sr)NiO₂ films which are similar in thickness but have no SrTiO₃ cap, Figure S5 of the supplemental reproduced as Figure R5 below.

Figure R5. Reproduction of Fig. S5 in the supplemental information. SIMS results from the as-grown (La,Sr)NiO₃ film corresponding to the reduced film shown in Fig. 2 of the original main text. As shown, the TiO₂⁻ intensity is effectively zero within the film despite the very thin (La,Sr)NiO₂ layer.

As is apparent in this figure, we observe a region with high NiO⁻/NiO₂⁻ intensities with effectively zero TiO₂⁻ intensity. That is, nearly all of the signal in the green region labeled (La,Sr)NiO₃ comes from the film. We therefore conclude that the layers are readily separable and that in the regions with high NiO⁻/NiO₂⁻ intensities are sensitive almost exclusively from the film. The SrTiO₃ caps of the superconducting films render the TiO₂⁻ peaks less informative in these samples, but the clear transitions from regions with no NiO₂⁻ to high NiO₂⁻, and back to very low NiO₂⁻ again demonstrate our ability to distinguish the film from the substrate and detect the hydrogen level within that layer.

Further, we have recently performed additional measurements (to be featured in an upcoming work) on *uncapped* (La, Sr)NiO₂ films which are either slightly thicker (8 nm) or much thicker (27 nm). As in the as-grown samples, the TiO₂⁻ intensity is near zero in the bulk of the film, denoted by the peaks NiO⁻ intensity in Fig. R6 below. Both of these films yield results that are consistent with the rest of this study.

Figure R6. (left) Additional SIMS results from an 8 nm thick (La,Sr)NiO₂ and (right) 27 nm thick (La,Sr)NiO₂ film, both without SrTiO₃ caps. As previously observed, the average H⁻ intensity in these films does not exceed 1.5× that of the substrate baseline

Having addressed the critical issue of “layer separability”, we can now define “depth resolution,” which is a measure of the apparent interface sharpness. The resolution will depend on instrumental background such as mass spectral overlap, the actual interface roughness, the uniformity of the sputtering crater, and variation in sputtering rate. We may approximate this using the time/distance required for a feature to decay from 84% of the maximum peak intensity to 16% ($\pm 1\sigma$ around the interface, defined as 50%). We will use a conservative estimate based on the NiO₂⁻ tail, yielding a resolution of approximately 5 nm - 6 nm.

Thus, we may be confident that the majority of the signal in the "film" region originates within the film, even in the thinnest samples shown in Fig. 2 - although we certainly agree that these are very challenging SIMS measurements. **We have the manuscript to better reflect this discussion. Specifically, the relevant section of the text now reads:**

“The change in molecular species intensity over time as the sample is sputtered results in a depth-resolved understanding of the chemical composition with depth, in which layers only a few nm thick may be readily separated.”

Further, we add the following discussion to the supplemental information:

“When discussing SIMS "resolution", the most important question is not the interface width but rather our ability to resolve the individual layers and identify their contributions to the signal. This is clearly achieved even in the thinnest ≈ 5 nm thick $\text{La}_{0.78}\text{Ca}_{0.22}\text{NiO}_2$ and $\text{La}_{0.8}\text{Sr}_{0.2}\text{NiO}_2$ superconducting layers, with clear onset and decay of the NiO_2^- signals which yield estimates of the $\pm 1\sigma$ interface widths of approximately 5 nm - 6 nm. We may therefore be confident that the vast majority of the signal in the regions designed "film" originates in the nickelate film, allowing the hydrogen content to be probed.”

In Figures 2e and 2f, there is a pronounced signal for NiO_2^- in the substrate and TiO_2^- in the film. Could the authors explain the origins of these strong signals

Response:

We thank the referee for bringing this issue to our attention. The long NiO_2^- tails are a classic example of cascade mixing effect in SIMS, where the high-energy analysis beam knocks atoms deeper into the film, smearing out the interface and causing a weak NiO_2^- signal to appear in a layer which hosts no Ni prior to the measurement [<https://doi.org/10.1002/sia.740040202>]. Mixing effects are most pronounced where the underlayer is lower in density than the overlayer, as in all of the nickelate/ SrTiO_3 layers studied in this work. We are therefore confident that the long tails represent a well-understood SIMS artifact rather than unexpected physics in the samples.

Cascade mixing effects may also play a role in the TiO_2^- signal observed in the nickelate in Figures 2e and 2f. Both of these films are superconducting films which were capped with a layer of SrTiO_3 , so that TiO_2^- is not expected to decay to zero within the thin film. This does not represent TiO_2^- within the film as-grown or after reduction. However, it should also be noted that the top interface of these films is known to be rougher due to the CaH_2 treatment and room-temperature deposition of a low-quality SrTiO_3 cap. This may also readily explain some of the TiO_2^- smearing. Here it is instructive to examine Fig. 5 of the main text, reproduced as R7, where STEM clearly shows separation between the SrTiO_3 cap and the nickelate film, while TiO_2^- signal in SIMS does not reach zero within the $(\text{Nd,Sr})\text{NiO}_2$ film.

Figure R7. Reproduced from Figure 5. (left) Cross-sectional STEM from a SrTiO₃-capped (Nd,Sr)NiO₂ film after reduction. (right) SIMS results from the same film. While the TiO₂⁻ peak does not decay to zero in the SIMS, the STEM clearly shows that the films are not intermixed with the cap or substrate.

We have added text in the supplemental information reflecting this discussion:

“On the other hand, we observed long NiO₂⁻ tails in the SrTiO₃ substrates, which may likely be attributed to cascade mixing effects induced by the analysis beam knocking atoms deeper into the film[52]. These effects may also account from some of the TiO₂⁻ signal observed in films capped by SrTiO₃, although the nickelate films are denser than the SrTiO₃ caps and these films are thin enough that the TiO₂⁻ is not expected to decay to zero.”

and

“In general, the SrTiO₃ substrate is expected to sputter more quickly than the higher-density films, although film quality also plays a role.”

and how the interfaces in the intensity profiles were determined?

Response:

As discussed above and in the supplemental information, standard convention describes the interface position between layer A and layer B as being the intensity which is halfway in between the TiO₂⁻ intensity of within layer A and layer B. We approximate the interface width as the time/distance required for a feature to decay from 84% of the maximum peak intensity to 16% ($\pm 1\sigma$ around the interface, defined as 50%).

Additionally, the superconducting La_{0.78}Ca_{0.22}NiO₂ and La_{0.8}Sr_{0.2}NiO₂ samples both have a thickness of 6 nm, yet the sputter times differ significantly (approximately 100s for LCNO and 300s for LSNO). What accounts for this discrepancy?

Response:

We appreciate this very astute question from the referee. Obtaining a high-quality ToF-SIMS dataset on such thin samples is extremely challenging. Additionally, SC nickelate samples are difficult to prepare and generally have small surface areas. Thus, only a few measurements may be performed with the destructive SIMS technique before the sample is unusable. We therefore were extremely cautious to avoid sputtering through the film too quickly, which is a particular danger when working with samples without predetermined sputtering rates.

Given the thinner SrTiO₃ cap protecting the superconducting La_{0.8}Sr_{0.2}NiO₂ film, we reduced the accelerating voltage of the sputtering beam prior to the measurement of this sample in order to ensure that we could identify the cap/film interface. The sputter conditions are 20 keV for LCNO and 10 keV for LSNO. Sputter rate is not exactly linear, so halving the voltage slightly more than doubles the sputter time. As a consequence, the frequency of the analysis beam per unit depth more than doubled, which

increased the extent of the mixing effect where the high energy analysis beam knocks the molecules deeper into the sample and smearing the interface. This is seen as the long NiO₂ tail in the SrTiO₃ for some measurements.

We apologize for the confusion arising by plotting depth as sputter time. As established, the sputter rate depends on the crystalline quality of the film, and was therefore different for as-grown and reduced films. As such, we increased or decreased the sputter beam current as necessary, especially due to the difference in thickness of the capping layers, but sometimes we did not have additional available area of the film to redo prior measurements at the exact same rate. We tried our best to plot samples of the same series using measurements taken at the same beam current, but did not have one setting which we used for all samples. The beam currents for the measurements mentioned are 20 keV for LCNO and 10 keV LSNO, respectively. Though sputter rate is not exactly linear with current, the higher current matches a faster rate. **We have clarified this in the figure caption so it is more obvious.**

When comparing results in the manuscript with those reported elsewhere, it is essential to provide references to support these comparisons, as noted around Line 197 on Page 3. Could the authors include the necessary citations to strengthen their discussion?

Response:

We appreciate the reviewers careful read of our paper. **We have added these in as References 32 and 33.**

Reviewer #2 (Remarks to the Author):

My field of expertise lies in TOF-SIMS. I found this work clearly showed that hydrogen incorporation was not required for superconductivity in these Nickelates. The authors were detailed in their explanations of how the SIMS data was collected and interpreted, and I see no need for revisions. I especially enjoyed seeing the attention to detail presented in supplementary sections on the SIMS data analysis. Also the details covered in supplementary section E are often not clearly stated in publications, even though I feel they should be. I congratulate the authors on their attention to detail in interpreting their SIMS data.

Response:

We thank the referee for their encouraging comments in support of publication.

Reviewer #3 (Remarks to the Author):

The present manuscript by Balakrishnan et al. aims to assess the level of importance of possible hydrogen intercalation into the lattice of square planar nickelates in relation to their superconducting properties. The tools used here are secondary ion mass spectrometry complemented with density

functional theory calculations. Acquiring accurate and reliable information about the role of topotactic hydrogen in these novel and highly debated systems would greatly advance our microscopic understanding of their superconductivity.

So far, however, hydrogen in the nickelates has remained a rather controversial issue. On one hand, several theoretical studies (Refs. 24-26 of the manuscript) consistently find it to be energetically unfavourable for hydrogen to enter the lattice of chemically hole doped nickelates. On the other hand, in a recent experimental work by Ding et al. (Ref. 23) superconductivity was observed only in samples with high measured concentrations of hydrogen.

In spite of using an identical experimental technique, the conclusions of the present work are radically different: the authors argue not only that hydrogen is irrelevant for the nickelate superconductivity but also that its concentration is always negligibly small in both superconducting and non-superconducting samples, irrespective of sample quality. These experimental findings seem to be in agreement with DFT calculations showing that hydrogen intercalation is unfavourable for the considered nickelate systems except LaNiO₂.

Response:

As a slight clarification, we would like to point out that we do not argue that the concentration is **always** negligibly small irrespective of sample quality. The DFT calculations which show that hydrogen intercalation is unfavorable assume an ideal square-planar crystal structure; they do not show that hydrogen cannot be intercalated into defects, such as the grain boundaries suggested by Pupal *et al.* (<https://doi.org/10.3389/fphy.2022.842578>). Thus, it is possible that hydrogen can be introduced with a high enough defect density. This is a focus of our future work, though outside of the scope of the current study.

As far as the theoretical part is concerned, I find the presented results scientifically sound within the capabilities of the adopted DFT approximations, which are clearly documented. The occasional differences with previously reported results for binding energies, in particular that of NdNiO₂, are attributed to using an improved DFT functional, but overall the qualitative trends regarding hydrogen intercalation are well reproduced.

What I find concerning, however, is the dramatic difference between the experimental data in the present work and in Ref. 23, both using same technique for measuring hydrogen content. If this technique allows for such substantial fluctuations in measuring the hydrogen content, then its reliability and scientific value are seriously questionable. The authors have to provide an explanation as to why their data should be considered more reliable than the work of Ding et al.

Response:

The referee raises a very serious and important issue, and we must be extremely clear what we are arguing. We believe that the experimental data shown by Ding *et al.* is a qualitatively accurate representation of the hydrogen distribution in the films used, but that their analysis and interpretation of that data is not correct. We do not believe that the differences between our studies point to a fundamental technique issue.

Specifically, **our claim is that in order to achieve a superconducting state in these layered nickelates, large amounts of incorporated hydrogen are not required.** The SIMS measurements in Ref. 23 do indeed show excess hydrogen in the top ≈ 10 nm of the films. However, these films are also vertically inhomogeneous, with a ≈ 5 nm thick low-hydrogen layer near the film/substrate interface. The low-hydrogen layer of the films could easily host the superconducting state.

Below, we will explain in detail why we disagree with the interpretation of Ref. 23, and how the experimental data is in some ways consistent between our study and theirs.

1. Differences between the ToF SIMS measurements

Our experimental technique is in fact not identical to that described in Ref. 23 (Ding *et al.*). While Ref. 23 used a Cs^+ sputtering beam, our work employed an Ar^+ cluster ion source as the sputtering mechanism. The choice of these ions used to bombard the surface can significantly change the chemical reactivity at the surface and the secondary ion yields, with Cs^+ known to dramatically lower the work function of the surface and enhance the secondary ion yield. This will naturally lead to significantly different absolute measurements of ion intensities, and it is for this reason that we refer frequently to the ratio between the film intensity and the substrate baseline intensity. This has been well established in literature on the technique, and we have discussed these subtleties in the supplementary information.

By considering relative intensity ratios instead of absolute intensities, the two approaches give fairly similar film/substrate H^- intensity ratios when significant incorporated hydrogen is present. Here we reference our own previous work using the same instrument to characterize H^- concentration in GdO_xH_y (<https://doi.org/10.1063/5.0128835>) and LaMnO_3H_x (<https://doi.org/10.1021/acs.nano.1c11065>). The large hydrogen content in these other systems has been corroborated by other experimental techniques. As discussed in our supplemental information, films with significant (10s of percent atomic ratio) hydrogen returned H^- intensities in the range of 20 \times - 300 \times the substrate baseline, with Ding *et al.* falling in the center of that range. Using film/substrate ratios to account for the difference in relative sensitivity factor between the different analysis beams thus suggests that the H^- intensities reported in Ref. 23 are reasonable for a sample containing significant excess hydrogen.

Consequently, we do not claim that our data is more reliable than that of Ref. 23. Rather, we believe our **interpretation** of our findings and theirs is more reliable. In particular, although we agree that their films contain significant hydrogen (though we see it is not uniformly distributed in the films), we contend that it has nothing to do with the superconductivity, as discussed below.

Figure R8. Intensity of H- peak normalized by the steady-state intensity in the substrate for three systems containing significant hydrogen, reproduced from Supplemental Figure S4. Blue: LaMnO_3H_x on SiO_2 (Ref. 32) Orange: GdO_xH_y on Si (Ref. 33) Grey range: Upper region of $(\text{Sr},\text{Nd})\text{NiO}_2$ on SrTiO_3 (Ref. 23)

2. Film Homogeneity

In Figure R9 below we reproduce the data from Figure 1(a) of Ref. 23, having downloaded the publicly available source data and followed their analysis based on the reported methodology. Using their methodology, we are able to reproduce their reported results for hydrogen concentration well.

Critically, we see that there are multiple film regions readily visible which were not discussed. These different regions are present in all films, including the as-grown film. For ease of visualization, we plot the data on both a log scale (left) and a linear scale (right). The surface, “Film A”, is either a transitory region from surface contamination, or a polycrystalline layer as found in (<https://doi.org/10.1063/5.0197304>). The hydrogen concentrations reported by ref. 23 represent only the values in the region labelled “Film B”, further from the substrate. In the region closest to the substrate, labelled “Film C”, the H-intensity of most reduced films is actually lower than the as-grown film. Further, the film/substrate H-intensity ratio in this region is similar to our samples, with $\frac{I_{\text{film}}}{I_{\text{substrate}}} \approx 5$ comparable to the $2\times\text{--}3\times$ limit for most of our samples.

Figure R9. Hydrogen concentration x of films from Ref. 23 following their reported methodology.

Ding *et al.* do not present any argument for why Film B is considered the superconducting region rather than other layers such as Film C. In fact, many in the field suspect that superconductivity in nickelate films mainly resides within higher-quality layers near the film/substrate interface. As film thickness increases, the upper portions of infinite-layer nickelate films are known to be more defective due to strain relaxation, which has been cited as a possible explanation for the increased challenge of achieving zero-resistance in thicker films. The films of Ref. 23 are 15 nm thick, very near previously reported upper limits. Thus, we speculate that the chemical inhomogeneity in these samples originates from structural inhomogeneity, as hydrogen incorporation may be more favorable by acting as a charge compensation mechanism for defects within the upper regions of the film (A/B).

Direct evidence of electrical inhomogeneity may be found by plotting the film resistivity from Figure 2a in Ref. 23, on a more conventional display, shown in Figure R10. Here it becomes apparent that while a broad range of films exhibit a resistive transition, only a single film reached the $\rho = 0$ state, despite some contradictory statements in the main text:

- A. “Zero resistivity is found within a very narrow H-doping window of $0.22 \leq x \leq 0.28$,”
- B. “In intermediate ranges (0.21, 0.22 and 0.28, 0.29), a superconducting-like resistivity transition occurs, but zero resistance cannot be achieved. For the x range of $0.24 < x < 0.26$, the zero-resistance superconducting state is obtained”
- C. “the superconducting samples ($x = 0.24, 0.26$ and 0.28)”
- D. “a superconducting dome with a narrow range of optimal H doping, $0.22 \leq x \leq 0.28$.”

Regardless, we note that the resistivity transitions shown below are broad and appear in many cases to have several distinct steps, consistent with what one might expect from inhomogeneous films. Thus, the resistivity data of ref. 23 suggests that the superconductivity does not span the entirety of the film.

Figure R10. Resistivity vs. temperature for (Sr,Nd)NiO₂ films reduced for different amounts of time, from Ref. 23. (left) Plotted on a log scale of temperature (right) Plotted on a linear scale, zoomed in around the superconducting transition.

Although Ding *et al.* identify this superconducting sample as having a hydrogen concentration of $x = 0.26$, we regard it as more likely that the superconducting state is hosted within the film region near the substrate, which has much lower hydrogen content, as this region is likely of higher crystalline quality. In other samples, the superconducting state may be discontinuously distributed, resulting in non-zero resistivity. In this scenario, the appearance of a dome-like behavior in Fig. 2(b) of Ref. 23 is simply due to a change in oxygen stoichiometry and film crystal quality as the annealing time is tuned. It has nothing to do with the hydrogen in the upper portion of the film, which is simply monotonic with annealing time and oxygen concentration.

3. SIMS Reliability vs. Interpretation and Quantification

We must again emphasize that we do not object to the SIMS data presented by Ding *et al.* in the sense that the experiment appears to have been performed correctly and it is clear that there is significant excess hydrogen in the upper half of the film. We do, however, disagree with the way they have extracted quantitative hydrogen content from their data. We note and appreciate Referee #2's support of our approach.

In Figure R9 above, we reproduced the calculations presented in Ref. 23 using the data supplied with the publication, and found that the hydrogen content is calculated assuming $\frac{n_{\text{NSNO}}}{n_{\text{Mica}}} = \frac{I_{\text{NSNO}}}{I_{\text{Mica}}}$ where n is the volumetric number density, and I is the intensity in counts/s. The concentration is then extracted using the unit cell volume of the NSNO film. By ignoring relative sensitivity factors (RSF) this assumes that the chemical and crystalline environments of superconducting NSNO and Mica are identical. They are not. In fact, the RSF changes significantly even between as-grown and reduced NSNO.

We highlight the issues with this simplistic calculation by applying the same approach to other elements in the compound (see Fig. R11) and showing that it gives unreasonable values. This calculation yields the completely unphysical conclusion that, upon just one minute of reduction, the Nd/Ni atomic ratio changes by a factor of ≈ 3 and then remains stable for the rest of the reduction process. Thus, as we explain in our paper, without an appropriate standard, quantification at this level is not possible.

Figure R11 From Ding. et al. extended data, upon reduction the Ni-intensity decreases by nearly 2× while the Nd-intensity increases by approximately 1.5×. Without accounting for relative sensitivity factors, this would correspond to a transformation from $Nd_{0.8}Sr_{0.2}NiO_3$ to approximately $Nd_{1.2}Sr_{0.2}Ni_{0.5}O_2$. This is obviously nonphysical.

Note also that the intensity of the Ni and Nd ions varies significantly across the film with regions which correspond to different hydrogen intensities (marked by dashed lines in Figure R11). This strengthens our earlier impression of inhomogeneity in the films.

To summarize, our interpretation of the SIMS data from Ref. 23 is that the as-grown film from Ding et al. contains three distinct regions. These layers could reasonably be:

Layer A: A surface scale layer caused by exposure to atmosphere

Layer B: A relaxed layer with defects and high hydrogen concentration

Layer C: A fully strained, low-defect superconducting layer closest to the substrate interface.

Due to the differences in chemical environment from the presence of various defects or structural differences, the RSF (and therefore the measured intensities of Nd, Ni, and O) are seen to vary in these three regions. The as-grown films contain some amount of hydrogen, possibly originating from the PLD target used for film growth (see Ref. 23 Fig. S3). As reduction occurs, both oxygen and hydrogen content in the films evolve, while hydrogen incorporation varies significantly between layers. The hydrogen concentrations reported in Ref. 23 represent only the values calculated for **Layer B**, while no evidence is presented localizing the superconductivity to this layer as opposed to the low-H **Layer C**.

For all these reasons, we argue that, while there is excess H in the upper regions of the films in Ref. 23, their raw SIMS data does not support the conclusion that an $x=0.25$ hydrogen doping level is required for superconductivity.

Reviewer #4 (Remarks to the Author):

I co-reviewed this manuscript with one of the reviewers who provided the listed reports.

We thank the referee for their efforts and their assessment included above.

REVIEWERS' COMMENTS

Reviewer #1 (Remarks to the Author):

The authors have responded carefully to our comments.

However, in the revised manuscript, the authors further modified the conclusions:

" Of course, this study does not demonstrate that superconductivity requires the absence of incorporated hydrogen. The results of Ding et al. clearly demonstrate that it is possible to realize the superconducting state in samples which contain significant hydrogen concentrations. This work instead indicates that many films appear resistant to hydrogen infiltration, and that superconductivity may be readily realized at very low hydrogen levels for which no theoretical evidence supports hydrogen-mediated superconductivity."

Overall, based on the SIMS study, the manuscript seems to be investigating whether significant concentrations of hydrogen are required for superconductivity, rather than hydrogen itself. This again does not fit well with the strong claim and title of the manuscript "Hydrogen is not necessary for superconductivity". This would diminish the significance of this work, although I believe this paper is worthy of publication and would attract interest from peers.

Reviewer #2 (Remarks to the Author):

This work focuses on SIMS analysis of hydrogen in topotactically reduced nickelates. My expertise lies in the field of SIMS. The authors rebuttal mostly concerned questions related to the details of SIMS measurements, quantification, and artifacts that may arise from measurements.

I feel the authors accurately addressed all of the other reviewer comments, correctly elaborating on the many nuances of SIMS measurements and data interpretation, and I have no comments to add. Also, there was an extended reply to why their data interpretation is so different from that of reference 23. In the section 3. SIMS Reliability vs. Interpretation and Quantification on page 16 of the rebuttal document, the authors show that in reference 23 the authors of Ref 23 used simple intensity ratios to incorrectly calculate the hydrogen content from SIMS data.

Frankly, ignoring the RSF for sims quantification is a giant red flag that the authors from reference 23 do not understand SIMS measurements and data interpretation, and I would not place any stock in their interpretation of SIMS data. Nevertheless, I feel the authors still presented a valid argument for why the films in Ref 23 could have had a higher measured SIMS intensity while also being superconducting, due to the multi-layer nature of the films in Ref 23.

Reviewer #3 (Remarks to the Author):

I am satisfied with the authors' responses and therefore recommend publication of this work in Nature Communications.

Reviewer #4 (Remarks to the Author):
